

# Active acoustic surveys reveal coastal fish community resistance to an environmental perturbation in South Florida

Benjamin M. Binder[1], Guillaume Rieucau[2], James V. Locascio[3], J. Christopher Taylor[4] and Kevin M. Boswell[5]

[1] Department of Biology / Marine Sciences Program / Florida International University, Florida International University, North Miami, FL, United States of America

[2] Louisiana Universities Marine Consortium, Chauvin, LA, United States of America

[3] Mote Marine Laboratory, Sarasota, FL, United States of America

[4] National Ocean Service, National Oceanic and Atmospheric Administration, Beaufort, NC, United States of America

[5] Department of Biology, Marine Sciences Program, Florida International University, North Miami, FL, United States of America

Corresponding author
Benjamin M. Binder,
bbind002@fiu.edu

## ABSTRACT

Coastal fish communities are under increasing levels of stress associated with climate variation and anthropogenic activities. However, the high degree of behavioral plasticity of many species within these communities allow them to cope with altered environmental conditions to some extent. Here, we combine meteorological information, data from hydroacoustic surveys, and recordings of goliath grouper sound production to examine the response of coastal fish communities to heavy rainfall events in South Florida, USA, that resulted in the release of excess storm water into surrounding estuaries and coastal waters. We observed a nearly 12,000% increase in water column acoustic backscatter following a heavy rainfall event of September 16th, 2015. Interestingly, estimates of school backscatter, a proxy for biomass, increased by 172% with the onset of the perturbation. Schooling fish density also increased by 182%, as did acoustically derived estimates of mean schooling fish length (21%). Following the perturbed period, school backscatter decreased by 406%, along with schooling density (272%), and mean schooling fish length (35%). Hydrophone and hydroacoustic data also revealed that goliath grouper (*Epinephelus itajara*) spawning aggregations were persistent in the region throughout the duration of the study and continued to exhibit courtship behavior during the perturbed period. Our observations demonstrate the high level of resistance common in coastal species but raises new questions regarding the threshold at which fish communities and reproductive activities are disrupted. As coastal land use continues to increase, and the effects of global climate change become more pronounced, more Before-After Control Impact (BACI) studies will provide improved insight into the overall response of nearshore communities to future perturbations and the cumulative effect of repeated perturbations over extended periods.

## INTRODUCTION

Coastal ecosystems are regularly exposed to various natural and anthropogenic stressors which can produce significant changes in community structure, behavior, and life history of coastal fish communities (*Wilson et al., 2006*; *Walther, 2010*; *Thom & Seidl, 2016*). In many areas, changes in perturbation regimes associated with increases in the severity or frequency of climate events over extended spatio-temporal scales are expected to have severe impacts on both aquatic and terrestrial ecosystems (*Wilson et al., 2006*; *Paddack et al., 2009*; *Nicholls & Cazenave, 2010*; *Knutson et al., 2010*; *Adam et al., 2014*). In particular, the increased frequency of perturbations related to human activity and climate change have been identified as major drivers of increasing biotic and abiotic stress in coastal zones. Most notable of these stresses include the urbanization of coastlines, recreational activity, and episodic pulses of freshwater run-off into nearshore systems that have seen the loss or deterioration of essential habitats (*i.e.,* port dredging and expansion efforts, recreational use of waterways and shorelines, etc.) (*Sime, 2005*; *Mallin, Johnson & Ensign, 2009*; *Hoegh-Guldberg & Bruno, 2010*; *Fabricius et al., 2014*; *Tilburg et al., 2015*). Broadly, perturbations are well-known to play pivotal roles in ecosystem dynamics (*Dornelas, 2010*), and while logistical limitations have historically hindered our ability to document the behavioral response of organisms to such events in near real-time, advancements in remote-sensing technology have provided opportunities to capture a range of behaviors throughout disturbance periods. Indeed, a growing body of knowledge has emerged to describe how estuarine, riverine, and coastal organisms respond to perturbations and extreme weather events (*Matich, Strickland & Heithaus, 2020*; *Massie et al., 2019*; *Strickland et al., 2019*). However, it remains unclear how offshore reef communities are linked to terrestrial perturbations, and how they will respond to future anthropogenically-mediated events influenced by changing climatic norms (*Dale et al., 2000*; *Hoegh-Guldberg & Bruno, 2010*).

South Florida regularly experiences these periodic environmental perturbations, such as heavy seasonal rainfall events that offer a natural experimental setting to examine the community response to rapidly changing environmental conditions. Following heavy rainfall events in South Florida, coastal ecosystems are often inundated with runoff from local urbanized areas and are also susceptible to significant freshwater inputs from regional watersheds. During these periods of elevated storm activity, flood mitigation activities scheduled by the South Florida Water Management District (SFWMD) and Army Corps of Engineers are implemented to alleviate stress on the Herbert Hoover Dike that surrounds Lake Okeechobee (Fig. 1), and to prevent wide-spread flooding in adjacent agricultural lands (*Zheng et al., 2016*). Recognizing the deleterious effects of untreated run-off into the estuary, the SFWMD uses stormwater treatment areas (STAs) throughout the region to mitigate the amount of untreated terrestrial material entering the estuary. Unfortunately, the magnitude and frequency of rainfall events between September and November 2015 exceeded the storage capacity of the adjacent STAs and required the controlled release of $5.79 \times 10^7$ m$^3$ of freshwater into the St. Lucie River estuary (personal communication SFWMD, DBHYDRO, Dec. 10, 2018). The influx drastically increased suspended sediment
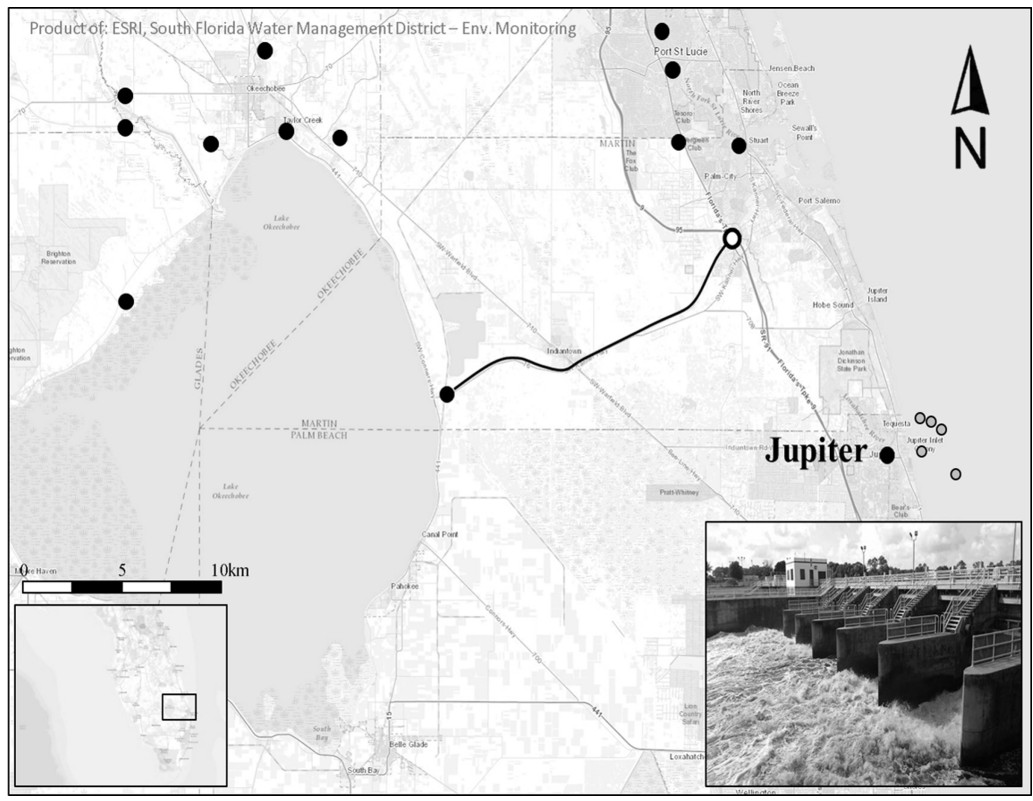

**Figure 1 Study region.** Closed black circles represent regional precipitation monitoring stations. The black line follows the St. Lucie Canal (C-44), the main waterway between Lake Okeechobee and the St. Lucie River Estuary. The white circle denotes the S-80 dam structure, the main drainage location for the St. Lucie and Okeechobee watersheds through the S-80 dam structure (inset picture credit: WPTV). Black outlined gray circles denote goliath grouper spawning sites and hydroacoustic sampling locations east of Jupiter, FL. Map credit: ESRI, South Florida Water Management District.

loads in coastal waters, and consequently diminished water clarity beyond 6 km from the shoreline for approximately one month (Binder, personal observation, 2015) This was also evidenced by satellite imagery from the time period that revealed increases in Chlorophyll-a concentrations in surface waters near the St. Lucie Estuary (Fig. 2).

Coastal environments frequently experience shifts in environmental conditions that have the potential to modify local community composition and structure. Thus, the organisms that persist in coastal systems are generally adapted to variable conditions, such as seasonal changes in temperature, tidal effects, and rapid changes in turbidity. Recent research has also depicted fish schooling behavior as a highly dynamic and plastic process, such that individual fish are capable of altering their behavior (*e.g.*, swimming faster and maintaining greater alignment with individuals in schools) in response to changes in local conditions to facilitate the transfer of information and favor survival (*Rieucau et al., 2016*). However, the ability to rapidly adjust to unexpected shifts in conditions may not extend to upper-trophic levels and has the potential to produce unpredictable changes in predator and prey dynamics (*Rogers, 1990*; *Syms & Jones, 2000*; *Leahy et al., 2011*; *Ponge, 2013*), disrupt

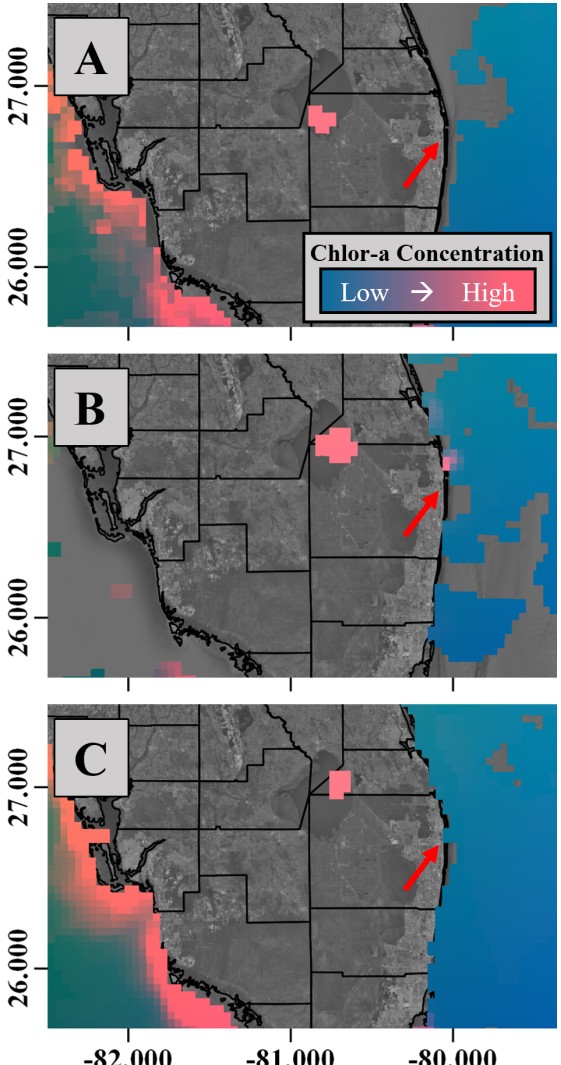

**Figure 2  Chlorophyll-a concentration through study period.** (A) Weekly chlorophyll-a concentration composites derived from satellite Imagery corresponding to the pre-perturbation period (September 6th, 2015); (B) the perturbed period (September 28th, 2015); and (C) the period following the perturbation (November 1st, 2015). Note that warm colors correspond to areas of high productivity. The red arrow references the St. Lucie estuary inlet.

important life history processes such as aggregative spawning (*Lewis, 1998*; *Nemeth, Sadovy de Mitcheson & Colin, 2012*), alter social behavior (*Berg & Northcote, 1985*), and decrease foraging success (*Gregory & Northcote, 1993*).

Among upper trophic level species of concern, the Atlantic goliath grouper (*Epinephelus itajara*) aggregates on South Florida reefs to spawn between August and November (*Koenig et al., 2016*) and may therefore be at particular risk. In this instance, the 2015 rainfall events occurred during the peak of their spawning season (*Koenig et al., 2016*), and though all life-history stages of goliath grouper spend a large portion of their time in nearshore habitats (*e.g.*, Florida Everglades and Florida Bay), it is unclear if rapid changes

in environmental conditions (*e.g.*, increase in turbidity, the passage of storms, or rapid influx of excess nutrients) affect their spawning behavior. There are numerous studies that have demonstrated persistent spawning behavior in coastal species experiencing intense storm activity (*Biggs, Lowerre-Barbieri & Erisman, 2018*; *Locascio & Mann, 2005*), but others have shown obvious shifts in activity that indicate a significant disruption in "day-to-day" behavior (*Bacheler et al., 2019*), that could include courtship and spawning activity. Further, even when temporary behavioral modifications are feasible, they are known to occur with poorly understood fitness trade-offs (*e.g.*, preference for sheltering over reproduction) that may lead to a significant decrease in ecosystem function, loss of biodiversity, and ultimately fish production (*Chabanet, Dufour & Galzin, 1995*; *Rooney & McCann, 2012*; *Wong & Candolin, 2015*).

In the case study presented here, we use regional meteorological data and hydroacoustic surveys to examine the relationship between rainfall events of September 2015 (13th–19th) and the subsequent changes in suspended materials in the water column (Fig. 3). Further, we investigate how this type of perturbation affects the morphological characteristics of fish schools (*e.g.*, length, area, thickness), relative abundance, size distributions, schooling fish density. Recordings from seasonally deployed hydrophone arrays were also used to characterize changes in goliath grouper sound production after the storm events and through the perturbed period.

## MATERIALS & METHODS

### Study region and data collection

Hydroacoustic surveys ($n = 31$) were conducted at five natural and artificial reef structures approximately 4–6 km east of Jupiter, Florida (N 26°56.650, W 80°04.370) at depths between 18–45 m (Fig. 1). The three artificial reef sites were deployed between 1989 and 1997 in 18–25 m of water, and ranged in size from 30–60 m in length. The Wreck Train is comprised of three high-relief artificial reefs (~5 m) surrounded by sand bottom, metal debris, and patches of exposed low-relief hard-bottom limestone rock covered in various benthic associated organisms (*i.e.*, sponges, algae, *etc.*). The MG111 reef is comprised of a barge covered in concrete and debris, that extends approximately 3–5 m into the water column. Immediately adjacent to MG111 (~25 m) a field of ~3 m standing (and fallen) concrete columns (~1 m diameter) spaced at random intervals over a sandy expanse, covering a roughly 100 m × 50 m space. The Sun Tug reef is a small artificial reef (a steel tugboat), approximately 20 m long, surrounded by sandy bottom. One notable artificial structure (~10 m × 5 m rectangular barge) lies near the Sun Tug reef, but neither extend more than 3–5 m into the water column. The two natural reefs are located at depths of 20 and 45 m and are characterized by a continuous high-relief (~3–5 m) reef slope and interspersed 3-dimensional "caverns" and overhanging structures. Active acoustic surveys were focused on prominent points along these features that are well-known to support GG aggregations. Indeed, all sites were selected based on their use as goliath grouper spawning aggregation sites (*Koenig et al., 2016*) and surveyed from September–November 2015, near peak new and full moons (*i.e.,* biweekly, when possible), to capture the peak of goliath

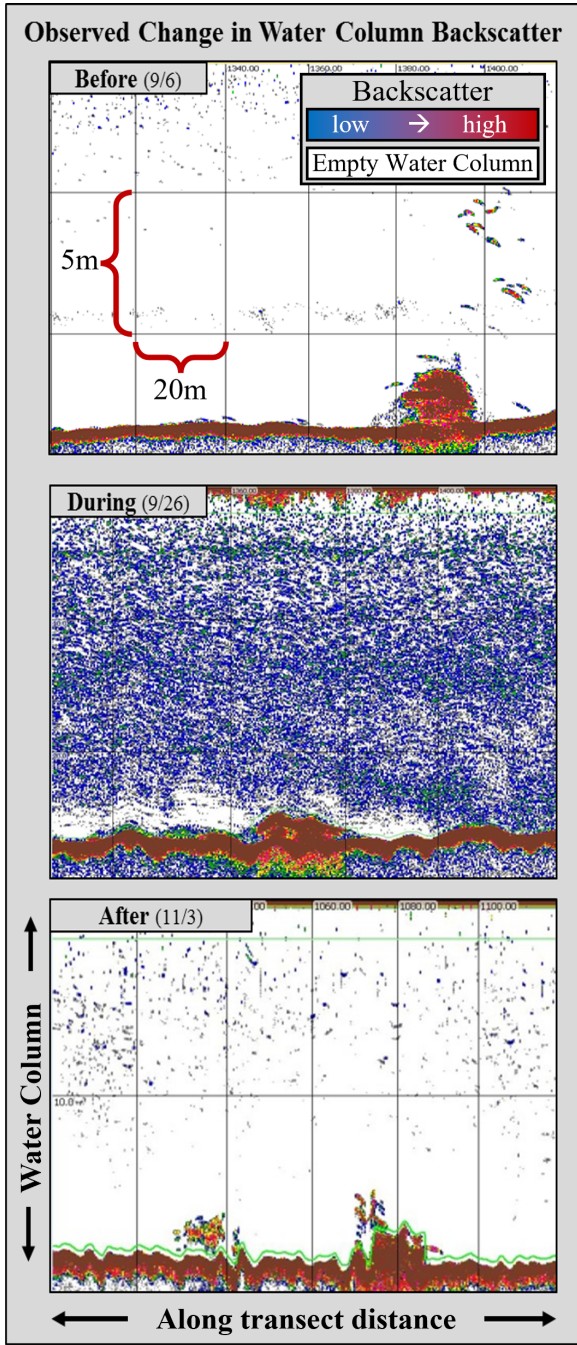

**Figure 3 Example echograms from the three distinct sampling periods.** (Before) Prior to the storm activity, water column backscatter was negligible, and individual fish targets were clearly observed in acoustic data. (During) Following the onset of storm activity and the subsequent freshwater releases, increased back scattering in the water column (blue pixilation in center panel) was observed. (After) Approximately one month after the storm-water control structures were closed, and estuarine flushing had concluded, water column backscattering returned to a "pre-disturbance" state.

grouper activity. Hydroacoustic surveys were conducted at approximately 2.5 m s$^{-1}$ and comprised of 8–12,600 m east–west linear transects at 25–30 m spacing that bisected the study reefs and surrounding habitat perpendicularly. Hydroacoustic data were collected with a calibrated Simrad EK60 120 kHz split-beam echosounder operating at 0.256 μs pulse duration with a 7° beam-angle. The transducer was deployed from a pole mount approximately 1 m below the surface. Standardized system calibration procedures were performed (*Demer et al., 2015*).

Passive acoustic recordings of goliath grouper sound production were made with calibrated DSG Acoustic Dataloggers (Loggerhead Instruments, Inc., Sarasota, FL, USA) at two of the study sites, from September 20 through November 29, 2015 (immediately following the passage of the storm period). Additional hydrophones were deployed, but they were lost or malfunctioned during the season. Acoustic data were recorded for 20 s every 5 min at 10 kHz sample rate. Sound pressure levels (SPL) for the 0–100 Hz frequency band were calculated for each .wav file as the mean SPL dB re: 1 μPa. Continuous recordings were not made because the technology (circa 2008) was not reliable beyond this level, though previous recordings using a similar duty cycle were sufficient to demonstrate well defined diel and seasonal patterns in Goliath grouper sound production (*i.e.,* 20 s/5 min) while preserving the limited amount of onboard flash memory (*Mann et al., 2009*).

A publicly available hydrological and meteorological dataset, DBHYDRO (http://www.sfwmd.gov/science-data/dbhydro), was queried for precipitation (cm), flow rate (m$^3$s$^{-1}$), and freshwater release timing corresponding to the study period. Daily precipitation (cm) from eleven monitoring stations in the Okeechobee and St. Lucie watersheds were selected to quantify rainfall in the region, and the St. Lucie Lock and Dam provided daily flow (m$^3$s$^{-1}$) into the St. Lucie Estuary along with the timing of dam openings (freshwater release events) (Fig. 1).

Exploratory dives were conducted prior to, and during the perturbed period, to confirm the presence of goliath grouper and describe the fish communities. The increase in turbidity following the rainfall events precluded standardized visual assessments, but divers did make qualitative assessments of the species present. Diver surveys consisted of paired point counts collected by independent observers at points on the respective study site. Divers completed a 10-minute visual survey, recording all species, their relative abundance, and estimated sizes. This included confirming that goliath grouper were present *via* direct observation and through the audible detection of their characteristic low-frequency vocalizations (aka 'booming'). Hook-and-line sampling from the survey vessel was also used to identify the schooling species observed in the water column following hydroacoustic surveys. Fishing surveys consisted of trolling over the study site with artificial baits (*i.e.,* 3 and 6″ cedar plugs and skirts) presented near the surface and midwater.

## Data processing

Echoview 8.0 (Sonar Data Pty. Ltd.) was used to process hydroacoustic data. An initial visual inspection of the raw data was conducted to identify and remove bad data and poor-quality data regions (*i.e.,* spike noise, rapid speed changes, abrupt turns). A bottom detection algorithm was then used with a 0.5 m back-step to eliminate reverberation
from the bottom, and a 2.0 m exclusion region was applied to the surface to eliminate the acoustic nearfield and artifacts from surface conditions (*e.g.*, bubble ringdown). Fish schools and individual fish targets were then flagged and isolated for exporting. Fish schools were identified with an automated detection algorithm within Echoview (minimum school height and minimum length = 1.00 m, minimum candidate height and minimum length = 0.20 m, maximum linking distance vertical and horizontal = 1.00 m). Point targets with target strength, TS >−50.0 dB (equivalent to standard length SL of >4.9 cm) were identified and tracked in Echoview using an alpha-beta tracking algorithm (*McCartney & Stubbs, 1971*; *Blackman, 1986*). The schools and target tracks produced by the algorithms were manually evaluated for errors, and incorrectly classified regions were removed from the final output. Water column backscatter data, excluding school and fish targets, were echo integrated in 5 m horizontal by 5 m vertical bins to derive estimates of the Nautical Area Scattering Coefficient (NASC; m2 nmi$^{-2}$). NASC estimates were used as an index of scattering in the water column attributed to detritus, plankton and flocculent matter (*Simmonds & MacLennan, 2005a*). Estimates of school NASC (*i.e.*, NASC measurements constrained to the fish school region), which is proportional to "acoustic biomass" or energy density (*Simmonds & MacLennan, 2005a*), were used to quantify changes in school biomass through the study period. The term "school NASC" is used hereafter as a proxy to describe changes in school biomass. Estimates of standard length (SL) were derived from point targets based on the relationship between TS and SL presented for a mixed assemblage of fish by *McCartney & Stubbs (1971)*; where

$$TS = 24.50 \times \log 10 (SL) - 66.84.$$

Point targets that were associated with schools and within 2.0 m of the school periphery (referred to hereafter as school adjacent fish targets) were used to estimate schooling fish length distributions (*Kloser & Horne, 2003*), while additional point targets within 100.00 m of the study reefs were used to generate non-schooling fish length distributions. The complete point target sampling distribution was decomposed into two separate distributions, capturing those targets suspected to be goliath grouper (>−35 dB, Binder et al., unpublished data collected 09-05-2017), and all other fish targets (>−50 dB and <−35 dB). A visual inspection of those data was then performed to confirm the presence of goliath grouper during the three sampling periods. Estimations of schooling fish density ($\rho$, fish m$^{-2}$) were derived from the area backscattering coefficient (s$_a$; m$^2$ m$^{-2}$); where

$$\rho = s_a / 10^{\left(\frac{TS}{10}\right)}$$

(*Maclennan, Fernandes & Dalen, 2002*).

Passive acoustic data were recorded at field sites from September 20 through November 29th, 2015. A Fast Fourier Transformation (FFT) of each 20 s .wav file was used to generate a power spectrum from which band sound pressure levels in 100 Hz wide bins were produced. Power spectra were calculated with a 1,024 point FFT resulting in a frequency resolution of 10 Hz, and a Hann window with 30% overlap was used to generate spectrograms for review. Passive acoustic data were analyzed with custom code written in MATLAB R2009B software (Mathworks, Inc., Natick, MA, USA), and spectrograms were reviewed in Raven
Pro 1.4 software using 100 randomly selected files from evening hours to confirm that GG were the source of sound productino. Nightly peaks in sound pressure levels in the 0–100 Hz band were indicative of goliath grouper courtship behavior as described by *Mann et al. (2009)*.

## DATA ANALYSIS

A Pearson product-moment correlation was performed to examine the relationship and the lag between rainfall and flow rate changes. A boot-strapped (1,000 iterations) trimmed-mean (10%) one-way analysis of variance (ANOVA) for heteroscedasticity, followed by a "lincon" multiple comparisons test was used to characterize the variation in water column NASC (*Wilcox, 2011*). The study period was then divided into three blocks (before, during, and after the perturbation). The metrics derived from the school detection and single target detection algorithms within Echoview (school area ($m^2$), school vertical distribution (m), school length (m), thickness (m), and school NASC), along with data pertaining to the acoustically derived mean length of schooling and non-schooling fish, were used to test for differences in school morphology (including schooling fish length), schooling fish density (fish $m^{-2}$), and non-schooling fish length.

School area and fish density were $\log_{10}$ transformed to meet the assumptions of normality and equal variance for a parametric ANOVA (*Cox, 2006*). School length and school thickness did not conform to the assumptions of a parametric ANOVA and were analyzed with a Kruskal-Wallis one-way ANOVA followed by a Dunn's post-hoc multiple comparison test. A k-sample Anderson-Darling Test performed on the remaining variables (school NASC, distance to seabed, school adjacent fish length, and individual fish length) found their distributions to differ significantly, and a boot-strapped (1,000 iterations) trimmed-mean (10%) one-way ANOVA for heteroscedasticity, followed by a lincon multiple comparisons test was used to test for differences between the perturbed periods (*Wilcox, 2011*). All analysis of hydroacoustic data, and presentation of DBHYDRO data were performed using R Statistical Software version 3.4.2 (*R Core Team, 2018*).

## RESULTS

### Environmental data synopsis

Between September and November 2015, $5.79 \times 10^7 m^3$ of freshwater was released into the St. Lucie River estuary. Approximately 34% ($1.95 \times 10^7 m^3$) of the total annual volume ($1.8 \times 10^9 m^3$) was released during the week of September 13-19, 2015, following 19 cm of precipitation over a three-day period (Fig. 4A). The peak flow of 98.97 $m^3 s^{-1}$ occurred on September 18, 2015, and was approximately three times higher than the average flowrate associated with freshwater releases (Fig. 4B). A positive correlation between rainfall and flow was found (Pearson's $r_{(74)} = 0.23$, $p < 0.05$) following a six-day lag. Field sampling intervals did not allow us to identify the lag between increased flow and changes in acoustic backscatter at the study sites (∼25 km south); however, mean water column backscatter (NASC) representing particles or debris, with fish targets and schools removed, was

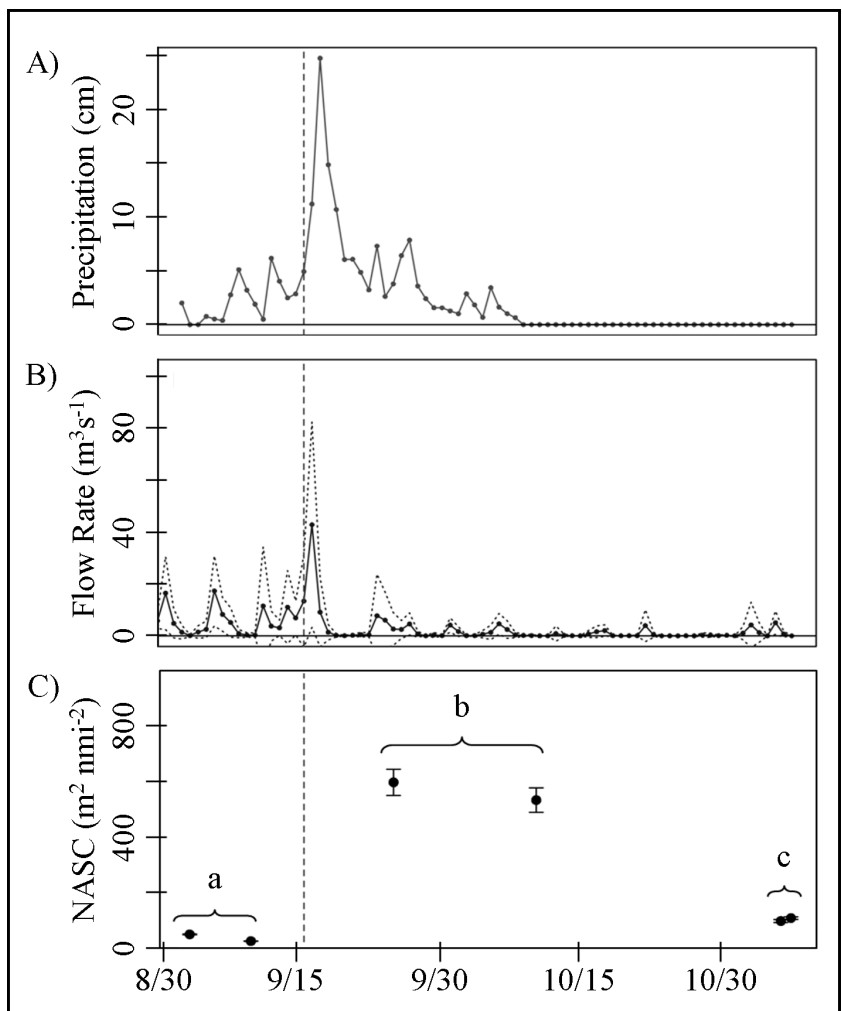

**Figure 4** **Environmental and active acoustic time series.** (A) Solid line denotes daily rainfall average (cm) from the region adjacent to Lake Okeechobee and St. Lucie Estuary (seen in Fig. 1). (B) The solid line denotes daily average flow rate through the St. Lucie dam structure (S-80) (seen in Fig. 1 inset). Dashed lines correspond to the standard deviation away from the mean associated with daily flow measurements. (C) Circles represent total water column acoustic backscattering (NASC; $m^2$ $nmi^{-2}$), minus scattering attributed to fish targets, with their associated standard error. Brackets and corresponding letters denote statistical significance and categorization of sample periods (a) before, (b) during, and (c) after the disturbance. The vertical lines represent the onset of the storm events beginning September 16th, 2015.

significantly higher during the perturbation, compared to both before and after (lincon: $p < 0.001$) (Fig. 4C).

## Fish community response to perturbation

School NASC, a proxy for biomass, was significantly different between all three periods (lincon: $p < 0.001$). Mean school NASC increased by 172% with the onset of the perturbation and decreased by 406% following the perturbation (Fig. 5A). Schooling fish density was also significantly higher during the perturbation compared to both before

and after (Tukey: $p < 0.005$), but density before and after were not significantly different from one another (Tukey: $p > 0.05$). Fish density increased by 182% with the onset of the perturbation and decreased by 272% following the perturbation (Fig. 5B). Standard length (SL) estimates of peripheral fish targets derived from target strength data were determined to be significantly different among all three periods (lincon: $p < 0.001$), however estimates were variable and increased by only 21% during the perturbation, decreasing by 35% after the event (Fig. 5C). Estimates of non-schooling fish length and school morphology (*i.e.,* school length, thickness, area, and vertical distribution) did not significantly vary throughout the study period ($p > 0.05$).

## Evidence of Goliath grouper occurrence

Diel patterns of sound production at goliath grouper spawning sites revealed nightly peaks ranging from approximately 90 to 110 dB SPL (re 1 µPa) through mid-October, a range consistent with goliath grouper courtship activity identified by *Mann et al. (2009)* (Fig. 6). The absence of diel spikes in sound production after mid-October, also indicated that goliath grouper were likely present but not exhibiting courtship behavior, which was further validated in decomposed kernel density estimations of target strength distributions that revealed persistent peaks at approximately −33 dB (*i.e.,* in the range assumed to be goliath grouper), though the study period. Also of note, kernel density plots reveal a bimodal distribution of fish targets ranging from −35 to −50 dB during the perturbed period. Peaks occurred at approximately −45 dB and −37 dB, characteristic of two dominant size classes of fish contributing to observed acoustic backscatter in the water column besides goliath grouper (Fig. 7).

## DISCUSSION

Our study demonstrates the direct relationship between terrestrial water management activities and the effects of large-scale water releases on coastal reefs in South Florida. Despite the occurrence of a significant increase in turbidity induced by heavy rainfall and the subsequent freshwater release from the St. Lucie Estuary, the local fish community exhibited a high level of resistance to the perturbation. Contrary to our predictions, the data revealed that the morphology and habitat use of reef-associated fish schools remained unchanged, whereas school NASC, density, and the mean schooling fish length increased during the perturbed period. Data from the hydrophones also indicated that goliath grouper continued their courtship behavior (*i.e.,* nightly chorusing) through the perturbed period, and we documented a natural cessation of vocalizations approaching the end of the spawning season. Notably, this is consistent with the observations of persistent spawning activity in various seatrout species (*Biggs, Lowerre-Barbieri & Erisman, 2018*; *Locascio & Mann, 2005*).

In addition to the apparent ability to resist change during the perturbed period, qualitative surveys by divers and trolling surveys documented an increase in the abundance of pelagic species, including little tunny (*Euthynnus alletratus*; hereafter referred to as bonito), round scad (*Decapterus punctatus*), mackerel scad (*Decapterus macarellus*), and spanish sardines (*Sardinella aurita*), that closely preceded or coincided with the onset of

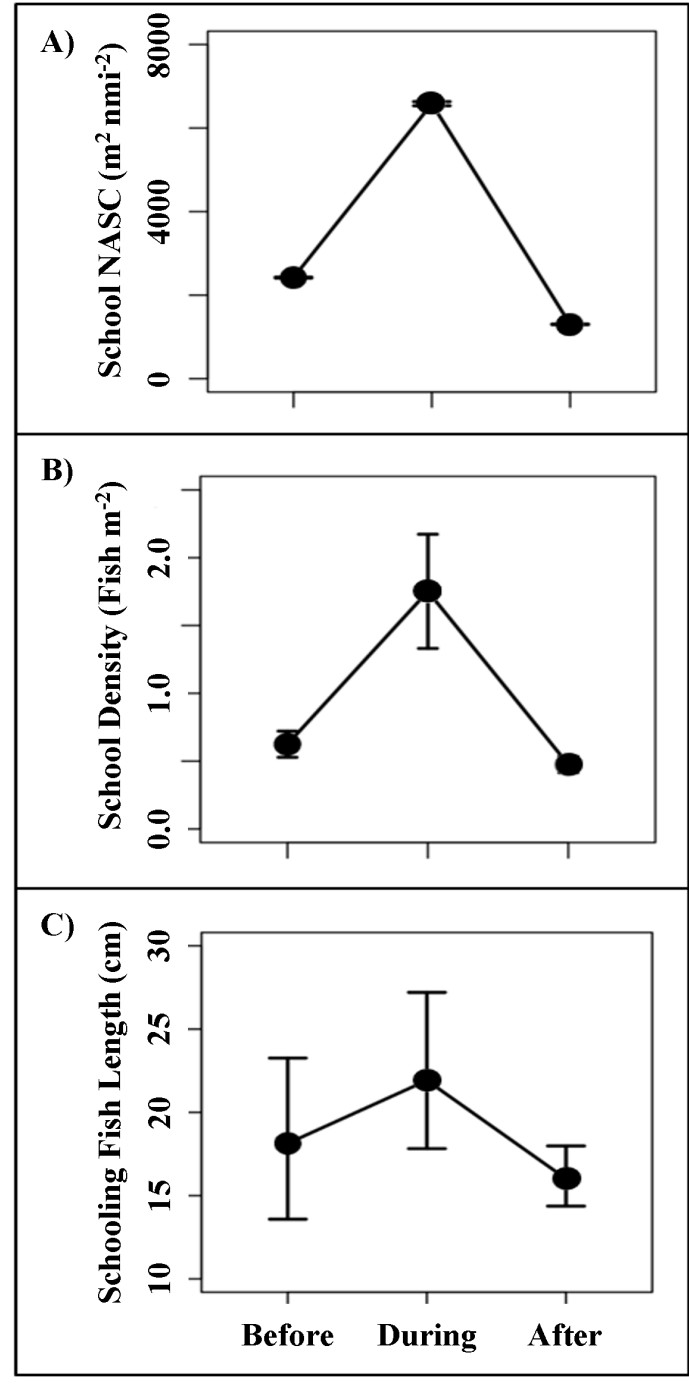

**Figure 5 Changes in school characteristics.** (A) NASC (m² nmi⁻²) estimates for fish schools from the three distinct study periods. (B) Estimates of areal schooling fish density ($\rho$, fish m⁻²) (C) Estimates of schooling fish size (standard length; SL (cm)), derived from TS measurements. Sizes were estimated using the target strength (dB m⁻¹) to length equation for mixed-species assemblage, where: $SL = 10^{(TS+66.84/24.5)}$.

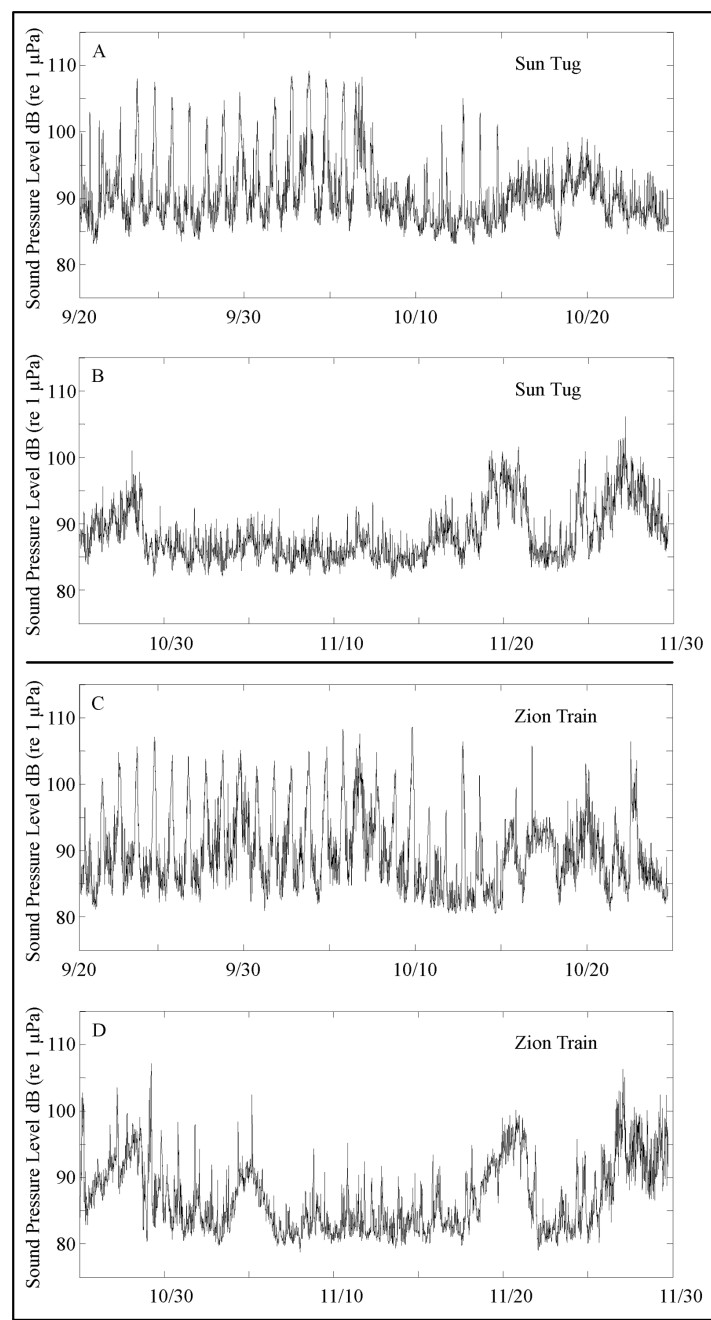

**Figure 6  Sound pressure level (dB re 1 μPa) through study period.** Band sound pressure levels of the 0–100 Hz frequency band recorded during September 20, through November 29th, 2015 at two Goliath grouper spawning aggregations sites near Jupiter, Florida; the Sun Tug (A and B) and the Zion Train (C and D). Nightly rises and falls in sound pressure levels associated with Goliath grouper courtship are evident between approximately 9/20 and 10/15. The new moon in October typically represents the end of the spawning season and this is reflected in the acoustic data.

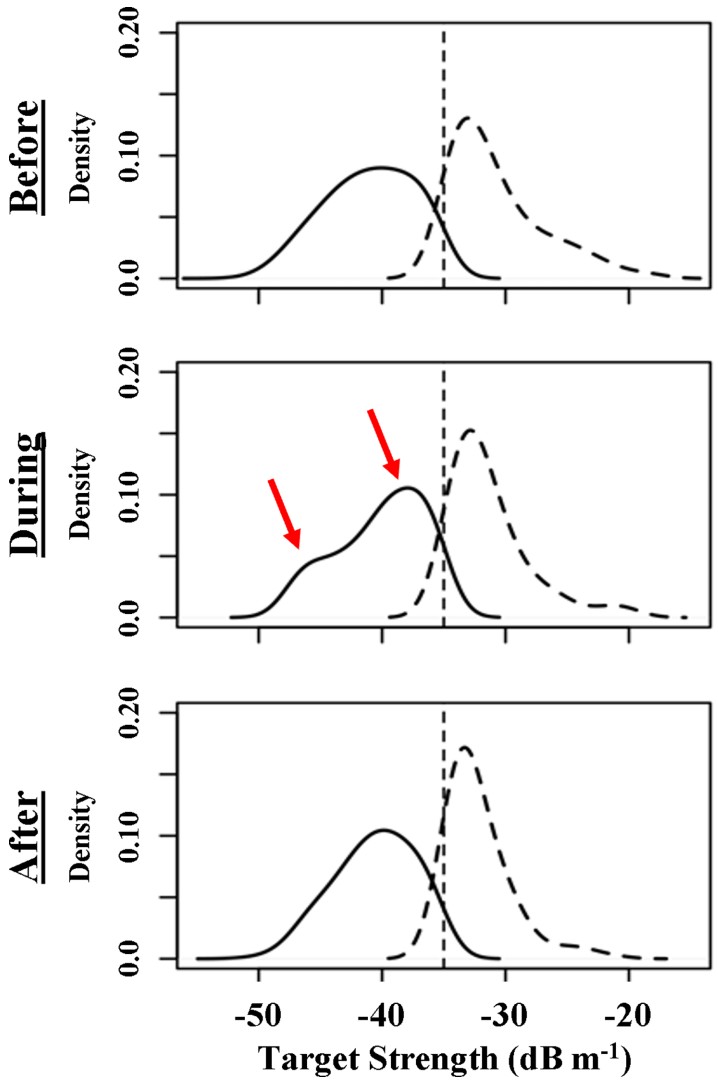

**Figure 7  Decomposed target strength distributions.** Decomposed target strength distributions for individually tracked fish targets based on kernel density estimation through the three periods. The complete sampling distribution was split at −35 dB, based on the assumption that targets > −35 dB were likely goliath grouper. TS estimates were aggregated into 1.5 dB bins. The vertical dotted line is used to simply denote the −35 dB division. Red arrows represent bimodal peaks suggestive of two distinct size classes observed during the disturbed period.

turbid conditions. Their arrival to the study area could explain the observed increase in schooling fish length, school NASC, and fish density that was observed during the disturbed period. In conjunction with the already present goliath grouper, the influx of bonito can explain the elevated estimates of school NASC, mean fish length, and school density.

Bonito, a mobile piscivorous species that forms large schools, were one of two numerically dominant fish species noted by divers compared to other common reef fish species present in the system before and during the perturbed period (Binder, personal observation, 2015). In addition to bonito, goliath grouper were present in high abundance

at some sites (5–50 individuals per site) by early September 2015, and likely continued to increase in abundance approaching the peak new moon spawning period (*Koenig et al., 2016*). Sound production at the study sites also confirmed that goliath grouper were present and continued to exhibit courtship behavior through the perturbed period (September 26–October 11), however it must be noted that the deployment of the hydrophones occurred three days after the passage of the storms. This limits our ability to make any definitive assessment of their response to the storm passage itself. The decrease in courtship associated sound production, following the new moon phase in mid-October, is consistent with the differential rates with which goliath grouper disperse at the end of the spawning season, and the persistence of small resident populations at the study sites throughout the year (*Koenig et al., 2016*). This was consistent with our observations of decreased school NASC and mean fish length after water column backscatter had decreased. Even at low abundance, goliath grouper are easily discernible as large single targets in hydroacoustic data when conspicuously present in the water column. As such, they have the potential to produce a significant positive shift in both metrics, due to their large swim-bladders and the associated acoustic response (*Love, 1971*; *Simmonds & MacLennan, 2005b*).

The combined influence of bonito and goliath grouper may explain the elevated school NASC and size estimates through the perturbed period, but their co-occurrence in the study area with several planktivorous fish species may also contribute to the observed changes. Round scad (*Decapterus punctatus*), mackerel scad (*Decapterus macarellus*), and spanish sardines (*Sardinella aurita*) are known to not only prey on goliath grouper eggs, but also use goliath grouper as a refuge from the piscivorous bonito (*Macieira et al., 2010*). The increase in planktivores abundance associated with the arrival of goliath grouper to the spawning sites, and possibly surplus food resources in the water column (*e.g.*, organic suspended material), likely attracted and sustained the bonito population through the perturbed period. It is reasonable to assume that the planktivores followed the natural dispersion of goliath grouper as the spawning season concluded in mid-October, and as suspended material in the water column decreased. Coincident with a decrease in prey biomass, bonito naturally dispersed in search of more abundant prey beyond our study area.

The arrival and departure of pelagic species likely explains a large portion of the observed changes in schooling structure, though behavioral changes in resident reef fish species cannot be dismissed as a factor that potentially contributed to our observations. Indeed, we recorded a decrease in the number of schools and non-schooling fish targets during the disturbance. While these two metrics were not considered reliable indicators of change due to their susceptibility to biases associated with data collection and processing methods, they could help to explain the observed changes in school density and school NASC. The net reduction in both schools and individuals detected through the perturbed period could be attributed to emigration away from the study sites, or a reduction in reef fish activity levels and/or a change in schooling structure that enabled them to avoid detection. Emigration away from regions affected by storms has been documented in numerous studies, with a range of species traveling meters to kilometers over hours to months to presumably evade inclement conditions (*Bacheler et al., 2019*; *Strickland et al.,*
*2019*). Alternatively, prey species (*e.g.*, members of the *Haemulidae* family) are known to decrease activity levels and form denser groups in response to increased predation risk and decreased sensory perception (*e.g.*, increased turbidity) (*Leahy et al., 2011*). Denser schools have the benefit of improved transmission of predator-based information through the collective group (*Rieucau et al., 2015*). The formation of denser groups that occupy less space also reduces the overall surface area of schools, limiting the points of vulnerability experienced by all school members. It is therefore possible that the increase in school density and NASC was a product of reef fish consolidating into denser units that remained close to the reef (*i.e.,* made fewer forays into open water). Our ability to justify this interpretation is potentially confounded by hydroacoustic data processing limitations, because schools swimming close to the reef can be difficult or impossible to detect due to occlusion by the acoustic deadzone (*Ona & Mitson, 1996*). However, that in and of itself is consistent with the reduction in schools and individuals observed, as we expect a proportion of schools and individuals did occur in the acoustic deadzone. From the remaining schools preserved for analysis, inclusive of the pelagic schools, the increases could in fact be partially attributed to the proposed behavioral changes exhibited by reef fish.

Coastal environments are highly dynamic systems that experience frequent shifts in environmental conditions, and the organisms that inhabit these locations must be resilient to seasonal changes in temperature, tidal effects, and various episodic events. This is especially true for goliath grouper (not excluding other coastal species), that inhabit inshore systems that experience frequent and acute shifts in conditions (*e.g.*, Florida Everglades, Florida Bay, *etc.*) throughout all life history stages, with no apparent consequences. If exposure to highly variable conditions was disadvantageous to their survival and performance, we would expect to see a shift in their distribution away from these habitats, though that is not the case. Indeed, consistent peaks in sound production associated with courtship through the perturbed period in our study provides evidence that the aggregations were present even after turbidity increased in response to the heavy rainfall event of September 16. However, despite their tolerance to turbid conditions, a direct correlation between sound production, courtship behavior, and active spawning has not been documented (*Mann et al., 2009*; *Koenig et al., 2016*). Previous studies have identified mixed response in overall activity, including spawning, with the onset of turbid conditions brought on by high-intensity storms, but the available information has focused on various smaller fish species (*i.e.,* those at relatively higher risk of predation) (*Leahy et al., 2011*; *Borner et al., 2015*), or species that consistently occur in turbid environments (*Bacheler et al., 2019*; *Biggs, Lowerre-Barbieri & Erisman, 2018*). Thus, it remains unclear whether the perturbation disrupted spawning during the peak of the spawning season, and further studies are needed to address this. However, considering their spawning season is concomitant to the wet season, when estuarine food resources are most readily available to juvenile goliath grouper, it is possible that these high-flow perturbations confer an unquantified advantage to dispersed larva (*Koenig et al., 2016*).

Our observations of increased water column acoustic backscatter following the storm events of September 2015 highlight the sustained response of coastal waters to inland water management activities (*i.e.,* the release of water from the water-control structure). The flow

of water into the estuary and out the adjacent inlets is a natural, well-documented process that estuarine and coastal species experience regularly, but modifications to the natural drainage patterns from the watersheds through man-made canal systems introduces a level of variability that these organisms may not be able to cope with. In the absence of man-made canals and water-control structures, unregulated flow through the aquifer would be distributed naturally, mitigating large pulses of water from being injected directly into the estuaries and coastal waters. Together, the decrease in light penetration, increase in siltation, nutrients, and terrestrially derived toxicant load associated with high flow from the estuary can have negative impacts on important aspects of estuarine and coastal ecosystem function (*Haunert, 1988*; *Sime, 2005*). As coastal land utilization continues to increase, and unpredictable high intensity storms become more frequent with changing climatic norms, the potential for large-scale environmental perturbations to disrupt ecosystem function and affect community dynamics will only increase (*Hoegh-Guldberg & Bruno, 2010*; *Walther, 2010*). While our data suggests that the reef-associated and pelagic fish communities resisted possible detrimental effects produced by the perturbation, and remained present throughout the period, it is unclear whether the conditions elicited any negative indirect impacts through behavioral or physiological effects on goliath grouper reproductive performance or larval recruitment to nursery habitats.

Resource managers are not unfamiliar with the effects of runoff on estuarine and coastal systems, though the extent of their relationship is not often clear and may be underrepresented in ecosystem-based management strategies. Indeed, the event described here has implications for managers involved in agricultural land use (*e.g.*, untreated fertilizer runoff), flood-mitigation, wetlands conservation, fisheries management, and myriad other issues across the State of Florida and analogous coastal areas. As we expect coastal land utilization to continue increasing into the future, along with a rise in storm intensity and frequency, a more comprehensive examination into the physiochemical processes that are associated with water column perturbations, and their effect on coastal fish communities is warranted. Lastly, more pre-emptive Before-After Control Impact (BACI) studies in areas that experience semi-predictable perturbations will provide improved insight into the overall response of nearshore communities to future events, informing the development of more effective ecosystem-based management strategies that ensure the sustainable use of coastal resources.

## ACKNOWLEDGEMENTS

We thank Jonathan Witmer for extensive assistance with data collection and processing, and Daniel Correa for assistance with R code development used to automate data processing. Additionally, we thank Danielle Morley, Dana Fisco, Pete Grasso, and Kirk Kilfoyle for their assistance in the field. This is contribution #1551 from the Institute of Environment at Florida International University.

### Funding

This work was supported by the Florida Department of Environmental Protection Coral Reef Conservation Program, the National Oceanic and Atmospheric Administration Coral Reef Conservation Program, the Cooperative Institute for Marine and Atmospheric Studies, and Marine Fisheries Initiative. The funders had no role in study design, data collection and analysis, decision to publish, or preparation of the manuscript.

### Grant Disclosures

The following grant information was disclosed by the authors:
Florida Department of Environmental Protection Coral Reef Conservation Program.
The National Oceanic and Atmospheric Administration Coral Reef Conservation Program.
The Cooperative Institute for Marine and Atmospheric Studies, and Marine Fisheries Initiative.

### Competing Interests

The authors declare there are no competing interests.

### Author Contributions

- Benjamin M. Binder conceived and designed the experiments, performed the experiments, analyzed the data, prepared figures and/or tables, authored or reviewed drafts of the article, and approved the final draft.
- Guillaume Rieucau conceived and designed the experiments, authored or reviewed drafts of the article, and approved the final draft.
- James V. Locascio performed the experiments, analyzed the data, authored or reviewed drafts of the article, and approved the final draft.
- J. Christopher Taylor conceived and designed the experiments, authored or reviewed drafts of the article, and approved the final draft.
- Kevin M. Boswell conceived and designed the experiments, performed the experiments, authored or reviewed drafts of the article, and approved the final draft.

### Field Study Permissions

The following information was supplied relating to field study approvals (*i.e.*, approving body and any reference numbers):

No permits required for conducting acoustic surveys offshore of Jupiter, Florida.

### Data Deposition

Active acoustic data were processed in Echoview. NASC data from water column evaluation, and school data from all surveys were compiled into a single file, and fish track data are similarly grouped. Environmental Data queried from DBHYDRO are also provided in the Supplemental Files.

## Supplemental Information

Supplemental information for this article can be found online at http://dx.doi.org/10.7717/peerj.14888#supplemental-information.

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
