# Peer review of "Active acoustic surveys reveal coastal fish community resistance to an environmental perturbation in South Florida"

_PeerJ, doi:10.7717/peerj.14888_

## Round 0.1 · original submission · Minor Revisions

I agree with the comments from the two reviewers of this article. The paper is well-written and technically sound. I don't believe there are any major edits or new analyses needed for a revision. Instead, I agree with the reviewers that there are a number of points in the methods and with some figures that should be addressed to improve reproducibility of the work. In addition, I would recommend the authors address a few points in the introduction from reviewer 1 with additional feedback on contextualizing the findings in the discussion from reviewer 2.

·

Basic reporting

The article passes this section. I would still recommend implementing the following:

• It is good practice to include the scientific name after the first mention of a study species, even in the abstract (Line 29).

• It would be helpful to elaborate on how perturbations are drivers of biotic and abiotic stresses. I recommend adding specific examples of biotic and abiotic impacts within Florida or areas with a similar ecosystem to Florida (Line 54-56).

• I appreciate mention the knowledge gap and including the reason the field lacks studies (Lines 63-68).

• Mentioning the flood mitigations strategies and how those strategies are challenged by increasingly severe rainfall events was well described and frames the importance of the study well (Lines 73-87).

• To improve readability and organization, I recommend breaking up the paragraph into how schooling fish communities are impacted/adjust to perturbation effects and how the goliath grouper is impacted (Lines 88 – 114)
o It would also be useful to include information on specifically how these severe and large rainfall events could negatively impact both fish communities and grouper. Are high turbidity levels an issue? It excess nutrient in the water from runoff a possible problem? By adding these details, it will better frame the importance for specifically looking at these two topics
o When talking about behavior modifications in grouper, specify examples of fitness tradeoffs (Lines 110 – 114)
o The information about behavior plasticity for both schooling communities and groupers and the need to understand the extent of this plasticity frames the study well

Experimental design

The study is well designed. However important details to make the study reproducible are missing. The paper fails this section, and I recommend the following changes:

• Reiterate the exact day(s) of the rainfall event (Line 116).

• Specify the exact dates or time interval between active acoustic surveys (twice weekly, bimonthly, etc.) (Lines 127-129).

• Specify why passive acoustic data was not recording continuously. Deployment appears short enough and sampling rate seems low enough where the continuous recording would be possible. Why was the data subset in this way? (Line 139)

• Specify parameters for power spectrum i.e.: FFT length, window,% overlap. Frequency resolution also needed (Line 194-195).

• Specify how passive acoustic data was analyzed in MATLAB (was it a packaged already published or custom code) (Line 195-196).

• Specify parameters to recreate spectrograms for reproducibility: i.e.: FFT length, window,% overlap. Frequency resolution also need. (Line 198-199).

• I would clarify if you examined spectrograms for all the data or just a subset of data (Line 199).

• Statistic methods are well described (Lines 202-223).

Validity of the findings

The article passes this section. I would still recommend implementing the following:

• Great job writing a clear and well-organized results section.

• For organizational flow, I recommend you include how you confirmed the increased presence of pelagic species here, rather than saying it later in the paragraph (Lines 274-278).
o I would also recommend specifying more clearly that this was a more qualitative observation that aligns with the hydroacoustic data. If it was more quantitative, please add those results to the result section.

• I appreciate you explaining the limitations of the interpretation of the hydroacoustic data (Lines 318-342).

• This sentence came across as misleading, as you did not determine that sound production increased (was elevated) over the perturbed period (although changes in the amplitude of sound production would be an interesting next study). The study only determined that sound production continued to persist over the perturbed period and did not analyze changes in sound production (Lines 350-353).

• The next steps still needed in this field were well described (Lines 393-397).

• Figure 3 needs to have axis labels and a legend.


Minor Comments:

• A legend would be helpful for Figure 2
o I would also recommend cropping the image to focus more on the areas of interest on the East Coast of Florida.

• It may also be interesting/helpful to include a spectrogram of the “boom” as an additional figure.

• If you make your data publicly available or share it, I recommend creating a Metadata file.

Additional comments

Reviewer Summary of Manuscript:

Binder et al. combines active acoustics and passive acoustics to understand the impact of intense rainfall in South Florida on coastal ecosystems. The study uses rainfall data to describe the weather event. To monitor changes in coastal fish communities during this perturbation event, the authors utilize active acoustics to assess responses of schooling fish and implement passive acoustics to monitor changes in the acoustic presence of an upper trophic level fish, the goliath grouper (Epinephelus itajara). The authors found an increase in biomass, schooling density, and mean schooling fish length during the perturbed period and a decrease following the perturbed period. The goliath grouper continued their spawning aggregations and acoustic courtship immediately following the rainfall event and after the perturbation period.

Overall, this is a very fascinating and well-designed study, and I applaud the authors’ use of both active and passive acoustics to assess the behavior of fish communities. With an increase extreme weather events inevitable, I think studies addressing these questions are very valuable for ecosystem conservation and management.

It is important that I add the caveat that my expertise is with passive acoustics, not active acoustics, so I am not able to critically review the active acoustic methods. However, I recommend this manuscript be accepted with revisions.

Reviewer 2 ·

Basic reporting

This article meets basic reporting standards: it is clear, professional, and well-justified.

Experimental design

This article meets experimental design standards: it is original research with well-defined objectives and appropriate methods.

Validity of the findings

This article meets finding validity standards: conclusions are supported by data and relevant literature.

Additional comments

My specific comments are found in the attached PDF.

Annotated reviews are not available for download in order to protect the identity of reviewers who chose to remain anonymous.

---

## Round 0.2 · accepted · Accept

Thank you for addressing the previous reviewer comments. I am happy with the current version and believe it is ready for publication.